# Extreme events driving year-to-year differences in gross primary productivity across the US

Alexander J. Turner[1], Philipp Köhler[2], Troy S. Magney[3], Christian Frankenberg[2,4], Inez Fung[5], and Ronald C. Cohen[5,6]

[1]Department of Atmospheric Sciences, University of Washington, Seattle, WA, 98195, USA
[2]Division of Geological and Planetary Sciences, California Institute of Technology, Pasadena, CA, 91226, USA.
[3]Department of Plant Sciences, University of California, Davis, CA, 95616, USA.
[4]Jet Propulsion Laboratory, California Institute of Technology, Pasadena, CA, 91109, USA.
[5]Department of Earth and Planetary Sciences, University of California, Berkeley, CA, 94720, USA.
[6]College of Chemistry, University of California, Berkeley, CA, 94720, USA.

**Correspondence:** Alexander J. Turner (turneraj@uw.edu)

**Abstract.** Solar-Induced chlorophyll Fluorescence (SIF) has previously been shown to strongly correlate with gross primary productivity (GPP), however this relationship has not yet been quantified for the recently launched TROPOspheric Monitoring Instrument (TROPOMI). Here we use a Gaussian mixture model to develop a parsimonious relationship between SIF from TROPOMI and GPP from flux towers across the conterminous United States (CONUS). The mixture model indicates the SIF-GPP relationship can be characterized by a linear model with two terms. We then estimate GPP across CONUS at 500-m spatial resolution over a 16-day moving window. We observe four extreme precipitation events that induce regional GPP anomalies: drought in west Texas, flooding in the midwestern US, drought in South Dakota, and drought in California. Taken together, these events account for 28% of the year-to-year GPP differences across CONUS. Despite these large regional anomalies, we find that CONUS GPP varies by less than 4% between 2018 and 2019.

## 1 Introduction

Terrestrial gross primary productivity (GPP) is the total amount of carbon dioxide ($CO_2$) assimilated by plants through photosynthesis and represents one of the main drivers of interannual variability in the global carbon cycle Le Quéré et al. (2018). As such, quantifying the spatiotemporal patterns of terrestrial GPP is critical to understanding how the carbon cycle will both respond to and influence climate. Work over the past decade has shown satellite measurements of solar-induced chlorophyll fluorescence (SIF) to correlate strongly with tower-based estimates of GPP (e.g., Frankenberg et al., 2011a; Yang et al., 2015; Sun et al., 2017; Turner et al., 2020; Wang et al., 2020) and are often used as a remote-sensing proxy for GPP.

This relationship between SIF and GPP is typically expressed through a pair of light use efficiency models Monteith (1972) that relate GPP and SIF to the absorbed photosynthetically active radiation (APAR):

$$\text{GPP} = \text{APAR} \times \Phi_{CO_2} \tag{1}$$

$$\text{SIF} = \text{APAR} \times \beta \Phi_F \tag{2}$$

where $\Phi_{CO_2}$ is the light use efficiency of $CO_2$ assimilation, $\Phi_F$ is the fluorescence yield, and $\beta$ is the probability of fluoresced photons escaping the canopy. Solving for APAR and substituting, we can rewrite GPP as:

$$GPP = \frac{\Phi_{CO_2}}{\beta \Phi_F} SIF. \tag{3}$$

The derivation follows from Lee et al. (2013), Guanter et al. (2014), Sun et al. (2017), and others.

This seemingly straightforward relationship between SIF and GPP has been widely used to infer GPP from measurements of SIF (e.g., Frankenberg et al., 2011a; Parazoo et al., 2014; Yang et al., 2015, 2017; Sun et al., 2017, 2018; Magney et al., 2019; Turner et al., 2020) with some work showing that SIF captures more variability in GPP than APAR alone (e.g., Yang et al., 2015, 2017; Magney et al., 2019). However, there is much complexity encapsulated in the first term of Eq. 3 ($\Phi_{CO_2}/\beta\Phi_F$). There is an ongoing debate about what *exactly* SIF is telling us about GPP (e.g., Joiner and Yoshida, 2020; Maguire et al., 2020; Dechant et al., 2020; He et al., 2020; Marrs et al., 2020) and the spatio-temporal scales at which SIF and GPP correlate well. A recent paper from Magney et al. (2020) presents a concise summary of the covariation between SIF and GPP at different spatio-temporal scales and how non-linear relationships at the leaf-scale often integrate to a linear response at the canopy-scale. This is due, in large part, to the fact that most of our satellite measurements occur near the middle of the day when the $\Phi_{CO_2}$-$\Phi_F$ response is more-or-less linear and the observed signal is integrated over many leaves.

Here we focus on the ecosystem-scale relationship between SIF and GPP, as that is the relevant observable scale from space-borne instruments. We begin by characterizing the relationship between instantaneous SIF from TROPOMI and half-hourly GPP from flux towers. Following this, we use this ecosystem-scale relationship to infer GPP at a spatial resolution of 500-m using TROPOMI SIF measurements and identify drivers of interannual variability in GPP. Previous work has identified effects and responses such as drought (e.g., Sun et al., 2015), flooding (Yin et al., 2020), and seasonal redistribution (Butterfield et al., 2020) as important factors controlling interannual variability in GPP.

## 2   Identifying distinct relationships between SIF and GPP

We build on our previous work (Turner et al., 2020) downscaling measurements of SIF to 500-m spatial resolution. Briefly, the TROPOspheric Monitoring Instrument (TROPOMI; Veefkind et al., 2012) is a nadir-viewing imaging spectrometer. TROPOMI has a 2,600 km swath with a nadir spatial resolution of 5.6 km along track and 3.5 km across track. Köhler et al. (2018) presented the first retrievals of SIF from TROPOMI. As in Turner et al. (2020), we apply a *post hoc* bias correction to ensure positivity of monthly average values as systematically negative SIF values are non-physical. We downscale individual TROPOMI scenes using the near-infrared reflectance of vegetation index ($NIR_v$) that was proposed by Badgley et al. (2017, 2019). We use the MCD43A4.006 (v06) MODIS NBAR reflectances Schaaf et al. (2002) to compute $NIR_v$. Two notable differences from Turner et al. (2020) are: 1) the analysis is extended to cover all of CONUS and 2) we now use a 16-day moving window, thus including a full orbit cycle in each averaging window to minimize effects due to viewing-illumination geometry and noise. Supplemental Fig. S3 shows the improvement when averaging to longer temporal windows with an $r$ of 0.66, 0.74, 0.79, and 0.82 for instantaneous, 8-day, 16-day, and 32-day temporal windows, respectively.

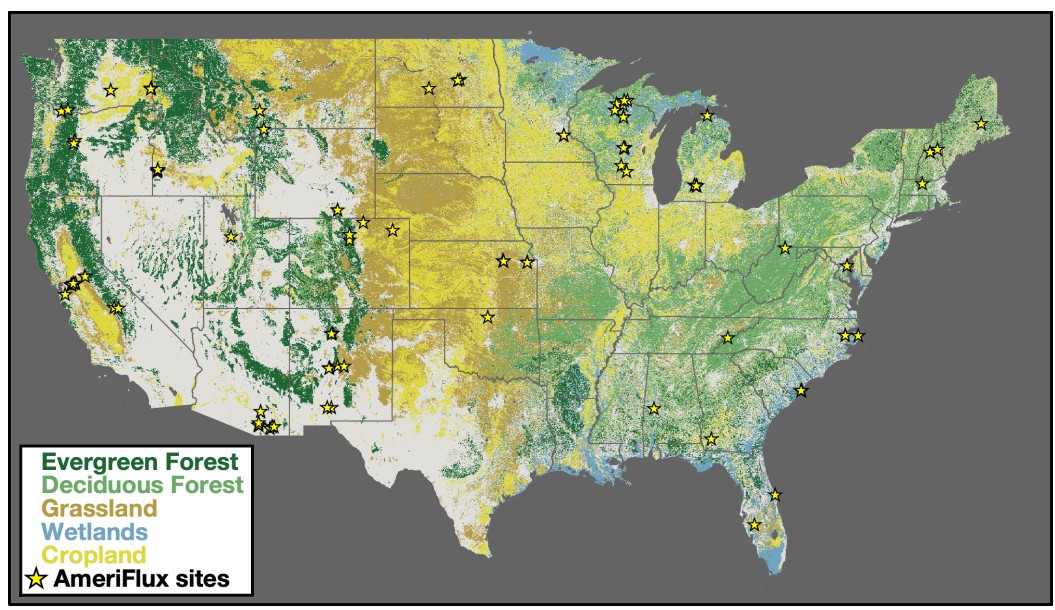

**Figure 1. Dominant landcover over conterminous United States (CONUS).** Colors show the dominant landcover over CONUS. Classification is based on the 2019 USDA CropScape database USDA (2018). Forests are shown in green croplands in yellow, and wetlands in blue. Location of 102 AmeriFlux sites used in this study are shown as yellow stars. See Table S1 for a list of all sites.

The extension to CONUS facilitates comparison of TROPOMI SIF retrievals to flux tower data over a more representative
set of ecosystems and robustly infer the SIF-GPP relationship. Specifically, there are 102 AmeriFlux sites Baldocchi et al.
(2001) within CONUS that reported data in 2018, 2019, or 2020 whereas Turner et al. (2020) only included 11 sites and did
not have data from forests. Figure 1 shows the location of these 102 AmeriFlux sites overlaid on the dominant landcover.
These eddy covariance sites provide a direct measure of net ecosystem exchange (NEE; $CO_2$ fluxes) Baldocchi et al. (1988).
We compute GPP at each site using nighttime measurements of NEE as a proxy for ecosystem respiration Reichstein et al.
(2005) to partition the NEE into respiration and GPP. The AmeriFlux sites used here cover 10 ecosystems as defined by the In-
ternational Geosphere-Biosphere Programme: evergreen needleleaf forest, deciduous broadleaf forest, mixed forest, grassland,
cropland, wetland, woody savanna, savanna, open shrubland, and closed shrubland. These are the classifications reported with
the AmeriFlux data as of July 2021 (https://ameriflux.lbl.gov).

We characterize the relationship between TROPOMI SIF and AmeriFlux GPP by plotting downscaled SIF observations
against daily GPP from the nearest AmeriFlux site (see Supplemental Figs. S1-S3). The TROPOMI overpass time varies over
the orbit cycle. Frankenberg et al. (2011b) presented a simple approach to compute a "daily corrected" SIF that accounts for
variations in overpass time, length of day, and solar zenith angle:

$$\overline{\text{SIF}}(x,y,t) = \text{SIF}(x,y,\tau_s) \frac{\int_{\tau_0}^{\tau_f} \cos[\text{SZA}(x,y,\tau)] \, d\tau}{\cos[\text{SZA}(x,y,\tau_s)]} \tag{4}$$

1    where $\text{SIF}(x, y, \tau_s)$ is the instantaneous SIF, SZA is the local solar zenith angle, $\tau_0$ is sunrise, $\tau_f$ is sunset, and $\tau_s$ is the

2    hour corresponding to the TROPOMI overpass time. We compare this daily corrected SIF against the daily GPP for each

AmeriFlux site. Specifically, the 7 steps we take here are: 1) construct a timeseries of daily GPP from each AmeriFlux site,

2) apply the *post hoc* bias correction to the TROPOMI SIF data, 3) compute the daily correction for TROPOMI SIF data,

4) downscale TROPOMI scenes to 500-m spatial resolution using MODIS NIR$_\text{v}$, 5) find all TROPOMI scenes that cover an

AmeriFlux site, 6) construct a timeseries of SIF observations from the 500-m grid cell that contains the AmeriFlux site, and 7)

regress coincident daily corrected TROPOMI SIF on daily AmeriFlux GPP with a bisquare regression. The bisquare regression

was chosen due to robustness against outliers. Additionally, we force the regression through the origin based on the physical

constraint that GPP should be zero if SIF is zero. We observe a linear relationship between SIF and GPP when plotted against

all ecosystems (Supplemental Figure S1) and when separated by ecosystem (Supplemental Figure S2). Notable exceptions are

closed shrubland, open shrubland, and savanna ecosystems where SIF explains less than 10% of the variability in GPP for

AmeriFlux sites in those ecosystems due, in part, to a low signal-to-noise ratio. These ecosystems typically have a small SIF

signal and the bright surfaces often result in a higher retrieval uncertainty. This combination of a small signal and high retrieval

uncertainty results in a low signal-to-noise ratio, complicating efforts to derive a robust relationship between SIF and GPP for

these ecosystems.

Many of the ecosystems exhibit a similar linear relationship between SIF and GPP, which begs the question: *"what ecosys-*

*tems have a distinct SIF-GPP relationship?"* To address this, we bootstrap the bisquare regression for each ecosystem 2000

18    times. The slopes from this bootstrap can be seen in Figure 2. The range of slopes vary from 13 to 18 $\left( \mu\text{mol}\,\text{m}^{-2}\,\text{s}^{-1} \right) / \left( \text{mW}\,\text{m}^{-2}\,\text{sr}^{-1}\,\text{s}^{-1} \right)$

with grasslands at the low end and evergreen needleleaf forests at the high end. We then use a two component Gaussian mixture

model (see, for example, Bishop, 2007) to identify clusters of ecosystems with a similar SIF-GPP relationship. The implemen-

tation of our Gaussian mixture model is adapted from Turner and Jacob (2015). Parameters of the mixture model are obtained

via an iterative expectation-maximization algorithm. A drawback of these mixture models is they often find local minima. To

address this, we repeat the fitting of the mixture model with multiple initializations and use simulated annealing to search for

a global minimum. We tested a range of mixture model sizes and found a mixture of two Gaussians to be the most robust.

Adding additional terms in the model resulted in Gaussians that did not have the largest weighting factor for any ecosystem.

This is because ecosystems like Woody Savanna and Deciduous Broadleaf have a large spread in their slope. As such, there

is a lot of uncertainty and the model does not find that they require a unique regression slope. The resulting mixture model is

overlaid on the histogram in Figure 2.

We observe a clustering of ecosystems with SIF-GPP relationships around 16.4 $\left( \mu\text{mol}\,\text{m}^{-2}\,\text{s}^{-1} \right) / \left( \text{mW}\,\text{m}^{-2}\,\text{sr}^{-1}\,\text{s}^{-1} \right)$. This

grouping is the dominant weighting term for wetlands, evergreen needleleaf forest, deciduous broadleaf forest, mixed forest,

cropland, and woody savanna. We refer to this cluster as the "Dominant Cluster" and assume that ecosystems not specifically

mentioned elsewhere will have a response that is similar to this primary cluster. The other component of the mixture model

corresponds to grasslands. Ecosystems not explicitly mentioned use the "Dominant Cluster" for scaling SIF to GPP. Table 1 lists

the SIF-GPP relationships for these two clusters. The uncertainty is the variance for the Gaussian for that particular cluster (see

Bishop, 2007; Turner and Jacob, 2015, for more on Gaussian mixture models). Previous work has also found unique SIF-GPP

**Table 1.** SIF-GPP relationships for different groupings.

| Cluster | SIF-GPP relationship[a] ($s_i$) |
|---|---|
| Dominant Cluster | $9.1 \pm 0.2$ |
| Grassland | $11.0 \pm 0.3$ |

[a]All SIF-GPP relationships have units of $\left(\mathrm{gC\,m^{-2}\,day^{-1}}\right) / \left(\mathrm{mW\,m^{-2}\,sr^{-1}\,nm^{-1}}\right)$. Uncertainty is the diagonal of the covariance matrix for the mixture model.

relationships between C3 and C4 plants using measurements from a tower including a non-linear response in C3 plants (He

et al., 2020), we examined this here using two AmeriFlux sites in corn fields and two in potato fields. We do observe potential

differences in the SIF-GPP relationship between these C3 and C4 systems (see Supplemental Figure S5). The difference in

SIF-GPP relationship for C3 and C4 systems seen here is also similar to what was observed using $\mathrm{NIR}_v$ Badgley et al. (2019).

These relationships can be used to reconstruct GPP from TROPOMI SIF as: $\mathrm{GPP} = \mathrm{SIF} \times \left(\sum_i f_i s_i\right)$ where $s_i$ is the SIF-GPP

relationship in Table 1 for the $i^{\mathrm{th}}$ cluster and $f_i$ is the fraction of a grid cell represented by that cluster.

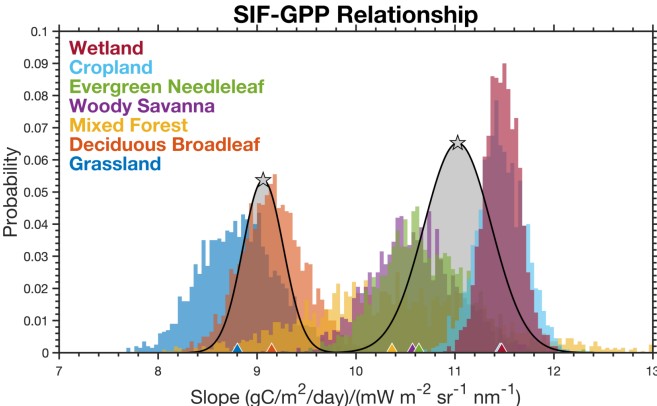

**Figure 2. Identifying distinct SIF-GPP relationships across ecosystems.** Histogram shows the distribution of slopes that map SIF to GPP using a bisquare regression and a 2000 member bootstrap. Colors denote the different ecosystems and triangles at the bottom show the mean for that ecosystem. Gray distributions are from a two-member Gaussian Mixture Model and the stars indicate the mean for that component.

7    TROPOMI is in low earth orbit and only observes a snapshot in time. The equatorial overpass time at nadir is 13:30 lo-

8  cal time. We compute a daily corrected SIF that accounts for variations in overpass time, length of day, and solar zenith

9  angle (Frankenberg et al., 2011b; Köhler et al., 2018):

$$\overline{\mathrm{SIF}}(x,y,t) = \mathrm{SIF}(x,y,\tau_s) \frac{\int_{\tau_0}^{\tau_f} \cos\left[\mathrm{SZA}(x,y,\tau)\right] \mathrm{d}\tau}{\cos\left[\mathrm{SZA}(x,y,\tau_s)\right]} \tag{5}$$

where $\text{SIF}(x, y, \tau_s)$ is the instantaneous SIF, SZA is the local solar zenith angle, $\tau_0$ is sunrise, $\tau_f$ is sunset, and $\tau_s$ is the hour

corresponding to the TROPOMI overpass time. We use this daily corrected SIF in conjunction with More formally, we scale

the instantaneous SIF by the ratio of the integral of the cosine of the solar zenith angle (SZA) over the day to $\cos(\text{SZA})$ from

the TROPOMI overpass time. Putting everything together, we estimate daily GPP from TROPOMI SIF observations as:

$$\text{GPP}(x, y, t) = \overline{\text{SIF}}(x, y, t) \cdot \gamma \sum_i s_i f_i(x, y) \tag{6}$$

where $\text{SIF}(x, y, t)$ is the 500-m downscaled SIF using a 16-day moving window, $\gamma$ is a unit conversion from $\mu$mol to gC, $s_i$ is

the SIF-GPP relationship inferred from comparison with AmeriFlux GPP (see Table 1), $f_i(x, y)$ is the fraction of the grid cell

represented by the $i^{\text{th}}$ cluster, SZA is the local solar zenith angle, $\tau_0$ is sunrise, $\tau_f$ is sunset, and $\tau_s$ is the hour corresponding to

the TROPOMI overpass time. We do not include information on cloud cover in our approach, this could potentially be included

in the future to account for diurnal variations in PAR.

Our estimate of GPP is proportional to SIF and the regression coefficients: $\text{GPP} \propto \overline{\text{SIF}} \cdot s_i$. As such, we propagate our

uncertainties in quadrature:

$$\sigma_{\text{GPP}} = \sqrt{\left( \frac{\partial \text{GPP}}{\partial \overline{\text{SIF}}} \sigma_{\overline{\text{SIF}}} \right)^2 + \sum_i \left( \frac{\partial \text{GPP}}{\partial s_i} \sigma_{s_i} \right)^2} \tag{7}$$

$$= \sqrt{\left( \sigma_{\overline{\text{SIF}}} \gamma \sum_i s_i f_i(x, y) \right)^2 + \sum_i \left( \overline{\text{SIF}}(x, y, t) \cdot \sigma_{s_i} \gamma s_i f_i(x, y) \right)^2}$$

where $\sigma_{\overline{\text{SIF}}}$ is the uncertainty in the daily corrected SIF and $\sigma_{s_i}$ is the uncertainty in the SIF-GPP relationship.

## 3   Drivers of interannual variations in US gross primary productivity

Figure 3 shows annual mean GPP across CONUS inferred from TROPOMI SIF measurements using Eq. 6. A number of

prominent features are visible such as the Central Valley of California, the Snake River Valley in Idaho, and the Adirondack

Mountains in upstate New York. California's Central Valley and Idaho's Snake River Valley are both major agricultural regions

in the western US (e.g., the Central Valley of Califoria accounts for more than 15% of irrigated land in the US). The Adirondack

Mountains are a roughly circular dome that rise above the surrounding lowlands, resulting in a shorter growing season and lower

annual mean GPP. This shortened growing season can be seen in an animation of GPP over CONUS (Supplemental Movie S1).

We observe substantial GPP across the eastern US (delineated here by 98°W) with annual mean values generally in excess

of 5 gC/m$^2$/day. This region accounts for less than half of the land but more than 70% of the annual GPP. This delineation in

GPP roughly coincides with the location of drylands in CONUS that are more sensitive to changes in precipitation as inferred

by measurements from the Global Precipitation Measurement (GPM) mission (specifically, we use the GPM_3IMERGDE.06

product); drylands are also projected to expand in future climate Yao et al. (2020). Most of the large year-to-year differences

occur in these western US drylands (see Fig. 3c), a notable exception being a negative GPP anomaly in 2019 relative to 2018

that extended across Illinois, Indiana, and Ohio. Here we highlight four precipitation-driven GPP anomalies, which taken

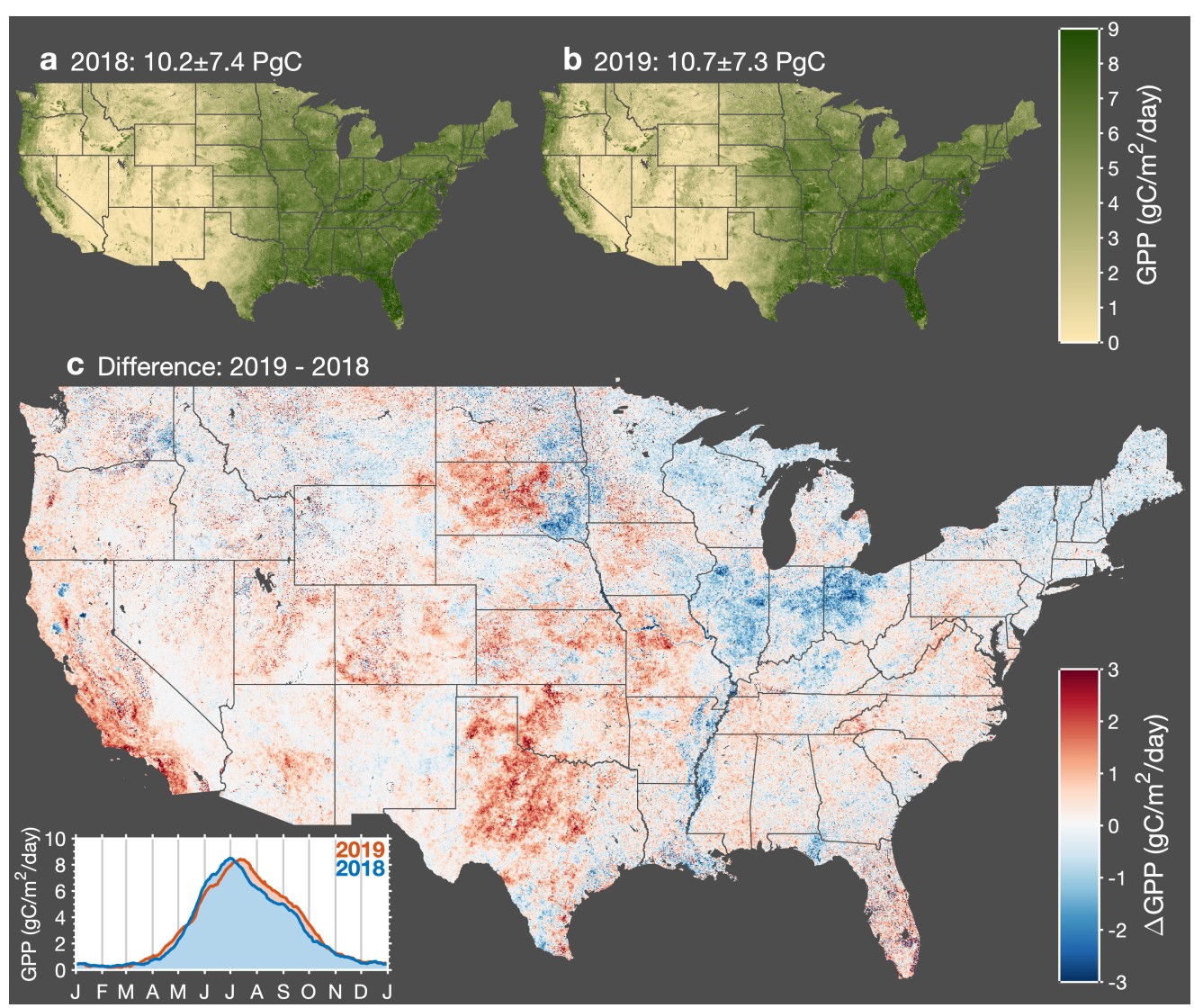

**Figure 3. Interannual variations in gross primary productivity across CONUS.** Map of annual mean GPP for 2018 (panel a) and 2019 (panel b). (Panel c) Map of the difference in annual mean GPP between 2019 and 2018. Red indicates higher GPP in 2019 and red indicates higher GPP in 2018. Inset in bottom left corner shows a timeseries of the average GPP across CONUS for 2018 and 2019.

together, account for 28% of the interannual GPP variability across the United States: 1) 2018 drought in west Texas, 2) 2019 midwestern crop flooding, 3) 2018 drought in South Dakota, and 4) 2018 drought in California. Figure 4 summarizes the interannual precipitation differences that we hypothesize are responsible for explaining these four GPP anomalies.

The largest positive GPP anomaly in 2019 relative to 2018 was observed across western Texas. This single event accounted for 11% of the year-to-year difference in GPP across CONUS with an annual GPP of $0.65 \pm 0.47$ PgC in 2018 and $0.76 \pm 0.45$ PgC in 2019. From Figure 4a, we observe 50% higher GPP in spring 2019 compared to spring 2018. This increase in GPP was driven by a lack of precipitation in spring 2018. The cumulative precipitation from October 2017 through June 2018 was 50% less than October 2018 through June 2019 (500 mm vs 1000 mm). The other notable difference between GPP in 2018 and 2019 was a second peak during fall 2018 that was not present in 2019. This second peak coincided with a series of precipitation events beginning in early September. This tight coupling between GPP and precipitation is expected for dryland systems such as west Texas (e.g., Smith et al., 2019). The seasonal GPP dynamics inferred from TROPOMI SIF are also present in the MODIS vegetation index $NIR_v$, albeit with slight differences in magnitude, implying convergent responses in SIF and $NIR_v$ for this ecosystem.

The second largest anomaly is the reduction in 2019 GPP relative to 2018 across midwestern crop areas (specifically Illinois, Indiana, and Ohio) that accounted for 7% of the year-to-year difference in CONUS GPP. The 2018 annual GPP was $0.70 \pm 0.12$ PgC and $0.63 \pm 0.14$ PgC in 2019. We observe a decrease in the maximum GPP between 2019 and 2018 as well as a two week delay in the timing of the maximum. This anomaly was highlighted in recent work from Yin et al. (2020) who attribute the anomaly to flooding in the midwestern US. The flooding delayed planting of crops by two weeks and resulted in decreased carbon uptake across the midwestern crop areas and Mississippi Alluvial Valley, where we also observe a negative anomaly in Figure 3c. Yin et al. (2020) provide a detailed discussion of these floods and their impacts on crop productivity.

South Dakota exhibits a dipole with positive anomalies in 2019 in the west and negative anomalies in the east, again relative to 2018. The 2018 annual GPP was $0.20 \pm 0.09$ PgC and $0.63 \pm 0.08$ PgC in 2019. The negative anomalies in the east are driven by the flooding events discussed above and in Yin et al. (2020). However, the positive anomaly in western portion of the state is the dominant term. This positive anomaly is driven by a series of summer precipitation events that served to extend the growing season across the western plains. From Figure 4c, we can see three precipitation events throughout the mid-to-late summer that coincide with pauses in senescence: mid-July, early August, and mid-September. As with Texas, this highlights the tight coupling between GPP and precipitation for dryland systems. In toto, these precipitation events served to increase statewide GPP in 2019 relative to 2018.

The final notable anomaly is California's positive GPP anomaly in 2019. The 2018 annual GPP was $0.27 \pm 0.24$ PgC and $0.33 \pm 0.26$ PgC in 2019. 2018 was a mild drought in California with $\sim$80% of the state being classified as abnormally dry; 2019 had 50% more precipitation during the water year than 2018 (Figure 4c). Two consequences of this drought in 2018 were: a delayed onset of photosynthesis and a mid-summer senescence. The onset of photosynthesis in 2018 coincided with a series of atmospheric rivers that delivered about a third of the total precipitation that year, indicating a water limitation up to that point. In contrast, 2019 had ample precipitation through the winter and we observe both an earlier onset of photosynthesis and an extension of the growing season into the fall. Evergreen forests are the main contributor to the SIF signal during the summer

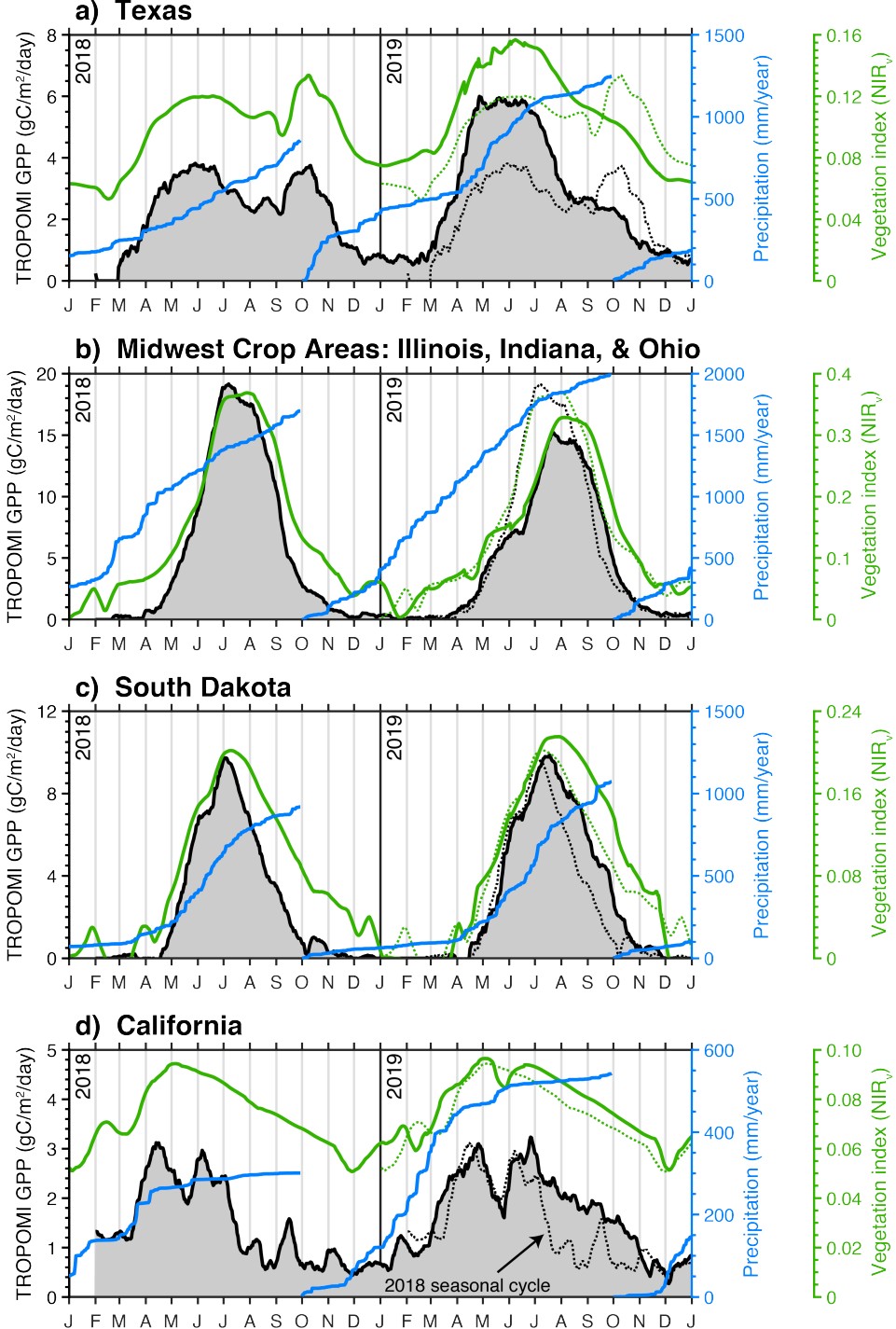

**Figure 4. Major drivers of interannual variability in CONUS GPP.** Black line shows the TROPOMI-derived GPP over Texas (a), the midwest crop region (b), South Dakota (c), and California (d). Blue line shows the cumulative precipitation over the water year as measured by the GPM satellite. Green line is NIR$_v$ from MODIS. Black and Green dotted lines are 2018 GPP and NIR$_v$ superimposed on the 2019 timeseries.

and fall Turner et al. (2020) and, as such, will be more sensitive to the accumulated precipitation. The spatial pattern of the differences in August-November GPP (Fig. S4) strongly correlate with evergreen forests.

In contrast to the anomalies presented earlier, the SIF-derived GPP and MODIS-based vegetation index ($NIR_v$) show divergent seasonal dynamics for California. $NIR_v$ shows small differences between 2018 and 2019 with a strong similarity to the 2019 SIF-derived GPP. The seasonality of $NIR_v$ is similar to that of the leaf area index (LAI) derived from MODIS (see Supplemental Figure 6), implying a biophysical signal. Vegetation indices derived from the red and NIR part of the spectrum estimate *photosynthetic capacity* provided optimal soil moisture, temperature, and PAR are known Sellers (1985). As such, this suggests that we observed a down-regulation of photosynthesis from evergreen forests in response to a water limitation during fall 2018, whereas these forests were close to photosynthetic capacity in fall 2019 resulting in a similar seasonality to 2018 and 2019 $NIR_v$. Sims et al. (2014) also report a low sensitivity of MODIS vegetation indices to drought stress in forests.

We additionally compare our GPP estimated from TROPOMI SIF to previous work developing gridded GPP products using machine learning. Specifically, the FLUXCOM initiative (http://www.fluxcom.org/; Jung et al., 2020) and FluxSat (Joiner and Yoshida, 2020) independently trained machine learning models to predict gridded-GPP from eddy covariance sites using remote sensing data (including MODIS). Figure 5 shows the CONUS seasonal cycle from both FLUXCOM, FluxSat, and TROPOMI. The seasonal cycles of GPP inferred from TROPOMI and FluxSat are generally in agreement with a similar magnitude while FLUXCOM predicts 35% less GPP. Additionally, the interannual variability in GPP over CONUS inferred from TROPOMI SIF is larger than what is predicted by either FLUXCOM or FluxSat, both of which show little interannual variability. The low interannual variability is particularly evident in FLUXCOM where we can see small spread in the variability from 2001-2017 (gray lines).

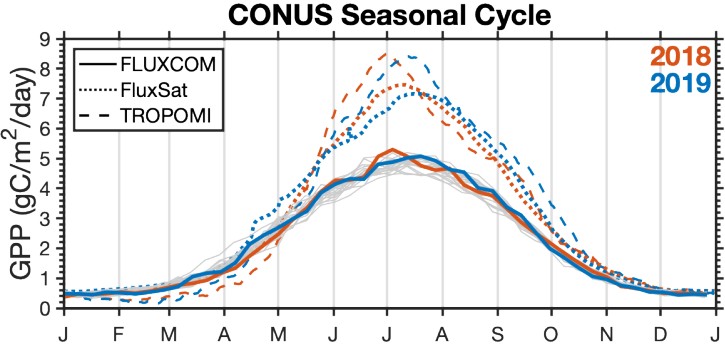

**Figure 5. Comparison of the seasonal cycle inferred from TROPOMI SIF to FLUXCOM and FluxSat.** Red lines indicate the 2018 seasonal cycle and blue lines indicate the 2019 seasonal cycle for TROPOMI (dashed lines), FluxSat (dotted lines), and FLUXCOM (solid lines). Thin gray lines are years 2001–2017 for FLUXCOM.

# 4 Conclusions

We have developed a parsimonious relationship between measurements of SIF from TROPOMI and GPP inferred from flux towers. This relationship allows for estimation of GPP directly from TROPOMI SIF measurements. We combine this SIF-GPP relationship with work downscaling TROPOMI data to 500-m spatial resolution to construct estimates of GPP across the conterminous United States in 2018 and 2019. We observe large regional anomalies that are driven by extreme precipitation events. Namely, west Texas, South Dakota, and California experienced droughts in 2018 while midwestern US crop areas (Illinois, Indiana, and Ohio) experienced flooding in 2019. Taken together, these four events account for 28% of the year-to-year variability in GPP across the conterminous United States. Despite these large regional anomalies, our estimate of US GPP varies by less than 4% between 2018 and 2019.

The impact of the west Texas drought, South Dakota drought, and midwestern flooding are observed in other remote-sensing measures of photosynthetic capacity such as $NIR_v$ while the California drought shows a divergent result using SIF; the divergent responses are driven by specific ecosystems such as evergreen forests. Our work suggests that SIF provides a measure of *photosynthetic activity* as opposed to *photosynthetic capacity*, and converge with other remote-sensing measures under non-stressed conditions. Future work investigating the response to extreme events in evergreen systems may provide additional insight into these divergent responses in remote-sensing measurements related to photosynthesis.

*Acknowledgements.* We are grateful to the team that has realized the TROPOMI instrument, consisting of the partnership between Airbus Defence and Space, KNMI, SRON, and TNO, commissioned by NSO and ESA. We acknowledge the following AmeriFlux sites for their data records: US-ALQ, US-ARM, US-Bi1, US-Bi2, US-CF1, US-CF2, US-CF3, US-CF4, US-CS1, US-CS2, US-CS3, US-EDN, US-GLE, US-Hn2, US-Hn3, US-Ho1, US-JRn, US-Jo2, US-KS3, US-Los, US-Me2, US-Me6, US-Men, US-Mpj, US-MtB, US-Myb, US-NC2, US-NC3, US-NC4, US-Rls, US-Rms, US-Ro4, US-Ro5, US-Ro6, US-Rwf, US-Rws, US-SRG, US-SRM, US-Seg, US-Ses, US-Sne, US-Snf, US-Syv, US-Ton, US-Tw1, US-Tw4, US-Tw5, US-UMd, US-Var, US-Vcm, US-Vcp, US-WCr, US-Whs, US-Wjs, US-Wkg, US-xAB, US-xBR, US-xCP, US-xDC, US-xDL, US-xHA, US-xJE, US-xJR, US-xKA, US-xKZ, US-xNG, US-xNQ, US-xRM, US-xSE, US-xSL, US-xSP, US-xSR, US-xST, US-xTE, US-xUK, US-xUN, US-xWD, US-xWR, US-xYE. In addition, funding for AmeriFlux data resources was provided by the U.S. Department of Energy's Office of Science. **Funding:** AJT was supported by the NASA Carbon Cycle Science Program (80HQTR21T0101) and as a Miller Fellow with the Miller Institute for Basic Research in Science at UC Berkeley. This research was funded by grants from the Koret Foundation and NASA 80NSSC19K0945 for support of the computational resources. Part of this research was funded by the NASA Carbon Cycle Science program (grant NNX17AE14G). TROPOMI SIF data generation by PK and CF is funded by the Earth Science U.S. Participating Investigator program (grant NNX15AH95G). TSM was supported through the Macrosystems Biology and NEON-Enabled Science program (DEB-579 1926090). This research used the Savio computational cluster resource provided by the Berkeley Research Computing program at the University of California, Berkeley (supported by the UC Berkeley Chancellor, Vice Chancellor for Research, and Chief Information Officer). **Author contributions:** AJT wrote the text with feedback from all authors. PK and CF performed the TROPOMI SIF retrievals. AJT downscaled the SIF data, conducted the AmeriFlux analysis, and drafted the figures. All authors contributed to the discussion and analysis. **Competing interests:** The authors declare no competing interests. **Data and materials**

1 **availability:** Daily gridded 500-m TROPOMI SIF and GPP data from February 1, 2018 through June 15, 2020 is temporarily available on

2 Google Drive here: "https://bit.ly/2GHEOOq", and will be uploaded to the ORNL DAAC at acceptance.

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
