# Peer review of "Extreme events driving year-to-year differences in gross primary productivity across the US"

_Biogeosciences, 2021_

## Author Comment (AC2)

**Response to Reviews:**

We thank the three Anonymous Reviewers for their comments on our manuscript.
* * *
**Reviewer #1 Comments:**

This study presents a GPP estimate over the conterminous US using TROPOMI SIF calibrated against eddy-covariance sites, with MODIS-based downscaling to 500 m spatial resolution. The methods used here are an extension of the previous work of Turner et al. (2020). This GPP estimate is then employed to examine interannual variations in GPP between 2018 and 2019, finding that differences between years are strongly impacted by four large precipitation-driven climate anomalies. This paper is well written, and I believe that this analysis is a significant scientific contribution and will be of interest to the readership of Biogeosciences. I feel that this paper is suitable for publication after generally minor revisions.

We appreciate the reviewers comments and feedback.

**Comments**

However, as a potential user of this datasets, I feel that the impact of this work would be significantly increased if a comparison with existing (particularly MODIS-based) GPP estimates was presented. Over the past several years, there have been a number of gridded GPP products developed, including the FLUXCOM GPP (Jung et al., 2020) and FluxSat v2 GPP (Joiner et al., 2020) that estimate GPP from MODIS data trained on eddy-covariance observations. A clear advantage of the MODIS-based GPP estimates is that they cover 2001-present, while the advantages of this TROPOMI-based GPP product are less clear. In particular, some questions that I have after reading this manuscript are: (1) Would the MODIS-based GPP estimates similarly find that 2018-2019 differences in GPP to be <4% with 28% of the variations explained by these four events? (2) Do MODIS-based GPP estimates show less IAV for forest ecosystems, as suggested by NIRv? (3) SIF and NIRv may underestimate drought-induced GPP reductions (e.g., He et al., 2020), what are the differences in drought-induced GPP reductions for TROPOMI-based and MODIS-based GPP? I encourage the authors to provide a comparison of the GPP estimated in this analysis with MODIS-based estimates.

This is an excellent suggestion from the reviewer. We now include a comparison of the FLUXCOM GPP and FluxSat GPP in addition to the comparison with NIRv and LAI.

Below is a figure showing the CONUS seasonal cycle for FLUXCOM (solid lines) and TROPOMI dashed lines. We note that TROPOMI finds a more abrupt seasonal cycle and more pronounced year-to-year differences than FLUXCOM. In response to the reviewer's question, it does seem that GPP estimates using MODIS data that are trained on eddy-covariance observations show less IAV across CONUS.

[Figure]

The manuscript does not provide much discussion of the uncertainties associated with these GPP estimates, and it is unclear if the GPP product provided with this analysis has associated uncertainties. I encourage the authors to include an uncertainty estimate with the data product and explain these uncertainties in the text. Presumably, an uncertainty estimate could be obtained from the uncertainty in the SIF-GPP regression.

Excellent point. We now include an uncertainty estimate by propagating the uncertainties in the SIF data and the SIF-GPP relationship:

Our estimate of GPP is proportional to SIF and the regression coefficients: $\text{GPP} \propto \overline{\text{SIF}} \cdot s_i$. As such, we propagate our uncertainties in quadrature:

$$\sigma_{\text{GPP}} = \sqrt{\left(\frac{\partial \text{GPP}}{\partial \overline{\text{SIF}}} \sigma_{\overline{\text{SIF}}}\right)^2 + \sum_i \left(\frac{\partial \text{GPP}}{\partial s_i} \sigma_{s_i}\right)^2} \qquad (7)$$

$$= \sqrt{\left(\sigma_{\overline{\text{SIF}}} \gamma \sum_i s_i f_i(x,y)\right)^2 + \sum_i \left(\overline{\text{SIF}}(x,y,t) \cdot \sigma_{s_i} \gamma s_i f_i(x,y)\right)^2}$$

where $\sigma_{\overline{\text{SIF}}}$ is the uncertainty in the daily corrected SIF and $\sigma_{s_i}$ is the uncertainty in the SIF-GPP relationship.

**Specific Comments**

*1)* P2 L18-20: Please re-word this sentence. Drought and flooding are drivers of IAV but seasonal redistribution is a response.

Done.

P2 L18: *"Previous work has identified effects and responses such as drought (e.g., Sun et al., 2015), flooding (Yin et al., 2020), and seasonal redistribution (Butterfield et al., 2020) as important factors controlling interannual variability in GPP."*

*2)* P2 L19: Check citation format. Many instances where "Author (year)" should be "(Author, year)"

We thank the reviewer for catching these issues. We have gone through all the references and corrected any LaTeX formatting issues.

*3)* P3 L6: "partitioned by the group operating the site". Is this always nighttime partitioning? Or does it vary between sites?

For consistency we no longer use GPP from the individual groups. We now compute GPP ourselves with nighttime partitioning at each site.

*4)* P3 L8: It is not clear what product is being used to define the IGBP product and version number (note that there are large changes in v6 of MODIS product).

This classification is based on the land type reported with the AmeriFlux data. We have clarified this in the text.

*5)* P4 L4: "small signal" is "small SIF signal"?

Done.

*6)* P4 L7-16: A little bit more detail could be given in this paragraph. Are you performing the cluster analysis on the 500 m gridcells?

We downscale each TROPOMI scene to 500-m spatial resolution and then find the pixels that contain AmeriFlux sites. We then regress SIF on GPP and perform a cluster analysis on the regression slopes.

We have elaborated on this discussion in the updated manuscript:

P4 L16: *"Many of the ecosystems exhibit a similar linear relationship between SIF and GPP, which begs the question: "what ecosystems have a distinct SIF-GPP relationship?" To address this, we bootstrap the bisquare regression for each ecosystem 2000 times. The slopes from this bootstrap can be seen in Figure 2. … We then use a two component Gaussian mixture model (see, for example, Bishop 2007) to identify clusters of ecosystems with a similar SIF-GPP relationship."*

*7)* P4 L15: "most robust" – how is this determined?

This was assessed through repeated analysis using mixture models of different sizes. We now include a supplemental table and figure for a Gaussian Mixture Model with 3, 4, and 5 Gaussians. A "hard clustering" of the data only results in 2 Gaussians that are the dominant term for an ecosystem see Supplemental Table SXX. This is because ecosystems like

Woody Savanna and Deciduous Broadleaf have a large spread in their regression slopes. As such, there is a lot of uncertainty and the mixture model finds that those ecosystems do not require unique regression slopes.

**8)** Figure 2: The colors for mixed forest and deciduous broadleaf are hard to distinguish and the histograms largely overlap. I suggest switching these to more contrasting colors.

The issue is that Mixed Forest has a large spread and actually lies behind the other histograms. Unfortunately, we tested a number of different color schemes and were unable to find one that drastically improve the visibility for mixed forest.

**9)** P7 L11: "(500 mm vs 1000 mm)" – I could not find where the precipitation dataset is described. Please check that the precipitation dataset is described and cited.

We have updated the text:

P6 L26: "as inferred by measurements from the Global Precipitation Measurement (GPM) mission (specifically, we use the GPM_3IMERGDE.06 product)"

**10)** P7 L30: "toto"

We believe this is grammatically correct. "In toto" is an adverb meaning "as a whole". This clause signifies how the various events come together to impact the total US GPP.

**Reviewer #2 Comments:**

Turner and others use SIF observations to estimate GPP across the U.S. and note that its interannual variability is driven in large part by extreme events. There are many interesting elements to the manuscript, but many aspects were difficult to follow and/or incompletely described which made it difficult to be confident in the results.

We appreciate the reviewer comments and have updated the manuscript to address them.

Referencing could be improved in many places (e.g. lines 19 & 22. On p. 2. Consecutive line numbering is so helpful, please don't re-start the line numbers on every page as a courtesy to reviewers.)

Unfortunately, this is the line numbering format is defined by the Copernicus LaTeX style sheet that accompanies Biogeosciences manuscripts. As such, this is the expected format for BGD manuscripts.

**Comments**

*1)* Figure 1: I have questions. Some of the sites shown are meant to study lakes or are in mountainous terrain, but it is hard to tell. Is there no list of sites used? Such a table would be extremely useful in the Supplement, and also to credit the data providers for their efforts in making the data available. At least one, if not more, appears to be a NEON site (anything starting with 'x'), which is fine but NEON should be credited. Ah, I see now that the sites are listed in the Acknowledgements. This is nice but a table would be more useful to the reader. (And US-Men is meant to study a lake. A table with ecosystem type and latitude/longitude would be helpful all around.)

*2)* Why were no eddy covariance sites in three of the four areas denoted as important for interannual variability (Texas, South Dakota, and Illinois/Indiana/Ohio) used? I understand that data are only intermittently available, but the Nebraska Mead sites may be a reasonable stand-in and the Indiana sites may be helpful.

We respond to comment #1 and #2 together as they are linked. We included all sites in the conterminous US that had data coincident with TROPOMI observations. There are many sites that did not have data available during the TROPOMI time period. TROPOMI launched in October 2017 and did not provide continuous data until 2018. We did not specifically preclude any sites. We now include a table in the supplement listing all the sites.

*3)* How was GPP estimated? Was this consistently using the nighttime (Reichstein et al., 2005 or similar) approach? If so was it consistently with the Reichstein algorithm? This uses a unique u* threshold every few months and respiration model parameters that change as a function of time. If these are interspersed with GPP estimates that use a different approach there will be differences in seasonality of GPP estimates due to methodological differences alone.

The reviewer is correct in identifying inconsistencies in the GPP methodology. The most obvious differences were in the methodologies used by the different groups operating the AmeriFlux sites. For example, some sites enforce a lower bound of zero for GPP whereas other sites do not (they allow variability with a mean of zero).

The choice made here was to use the GPP that has been partitioned by the group operating the site (if available) as they are most familiar with their site. If the group operating the site does not provide GPP then we compute it using nighttime NEE to obtain respiration. Again, our logic for this was that the groups operating the sites know them best and we should use their reported GPP.

In response to the reviewer comments we now compute GPP ourselves for all sites using nighttime NEE to partition GPP and respiration (Reichstein et al., 2005). We have changed this methodology to avoid inconsistencies in the computation of GPP.

P3 L6: *"We compute GPP at each site using nighttime measurements of NEE as a proxy for ecosystem respiration (Reichstein et al., 2005) to partition the NEE into respiration and GPP."*

*4)* Figure 2: I'm not entirely convinced that the 'dominant cluster' approach is useful or even necessary given that the peaks of the pdfs vary between about 15 (mixed forest) and 17 (evergreen needleleaf forest) and that estimating these land cover types using MODIS is rather well-established.

As stated in the manuscript, the question addressed by this analysis is: *"what ecosystems have a distinct SIF-GPP relationship?"* This cluster analysis across a diverse set of eddy flux sites allows us to identify unique relationships that arise between TROPOMI SIF and eddy flux GPP. This is useful because it then allows other researchers to estimate GPP from SIF even if their region does not include eddy flux towers in all ecosystems. This question was motivated by our own previous work in California (Turner et al., 2020). In that work we did not have eddy flux data in forest systems. As such, we were unable to say anything about those systems.

*5)* In Table 1, is the value behind the +/- sign the standard error of the mean or the standard deviation? (I'm assuming its not the variance). The table legend was not described in very much detail.

The +/- is the diagonal of the covariance matrix for that Gaussian as inferred from the Gaussian Mixture Model. We now add citations with more information on GMMs:

P4 L34: *"The uncertainty is the variance for the Gaussian for that particular cluster (see Bishop, 2007; Turner and Jacob, (2015); for more on Gaussian mixture models)."*

*6)* Per the previous comment, the structure of the manuscript was a bit bewildering. It was difficult to determine how the methodology took place because methods were interspersed throughout the manuscript starting on Page 1 with the equations. For example, I appreciate that uncertainty is (mostly) noted about carbon flux estimates but how is this uncertainty determined? Is the bootstrapping approach used to determine the uncertainty of the SIF-GPP relationship which is then propagated? Is uncertainty owing from the 16-day moving window used?

Following Reviewer #1's suggestion, we now include a paragraph discussing how we propagate uncertainty in our work:

P6 L11:

Our estimate of GPP is proportional to SIF and the regression coefficients: $\mathrm{GPP} \propto \overline{\mathrm{SIF}} \cdot s_i$. As such, we propagate our uncertainties in quadrature:

$$\quad \sigma_{\mathrm{GPP}} = \sqrt{\left(\frac{\partial \mathrm{GPP}}{\partial \overline{\mathrm{SIF}}} \sigma_{\overline{\mathrm{SIF}}}\right)^2 + \sum_i \left(\frac{\partial \mathrm{GPP}}{\partial s_i} \sigma_{s_i}\right)^2} \tag{7}$$

$$\qquad = \sqrt{\left(\sigma_{\overline{\mathrm{SIF}}} \gamma \sum_i s_i f_i(x,y)\right)^2 + \sum_i \left(\overline{\mathrm{SIF}}(x,y,t) \cdot \sigma_{s_i} \gamma s_i f_i(x,y)\right)^2}$$

where $\sigma_{\overline{\mathrm{SIF}}}$ is the uncertainty in the daily corrected SIF and $\sigma_{s_i}$ is the uncertainty in the SIF-GPP relationship.

**7)** Per the previous comment again, extreme events often happen quickly by definition. Is a 16-day moving window sufficient to fully describe how flooding of flack drought for example impacts ecosystems? I appreciate that the proposed precipitation mechanisms are described as 'hypothesized' and the notion that precipitation is the culprit makes great sense, but the devil might be in the details, which were not described again in sufficient detail.

Figure 4d demonstrates how our approach is sufficient for describing such events. In particular, we point the reviewer to the event that occurred on March 22, 2018. An atmospheric river delivered a large amount of precipitation to the state of California. This can be seen as the abrupt jump in the cumulative precipitation in 2018. We observe a coincident jump in our GPP estimate that is derived from TROPOMI SIF. As such, this serves as evidence that our approach is able to capture the types of phenomenon discussed here.

[Figure]

**8)** 30: GPP does not scale linearly with PAR; the GPP/PAR relationship will usually be saturated at 1:30 pm during the growing season. How does this impact equation 4?

We have made a slight change to the way the SIF:GPP relationship is inferred that should address this. We now compute a daily corrected SIF (see, for example, Frankenberg et al., 2011; Koehler et al., 2018) that corrects for variations in overpass time, length of day, and SZA. We then compare this daily corrected SIF with the daily integrated GPP at each AmeriFlux site. The relationship is now between daily corrected SIF and daily integrated SIF. As such, we no longer need to assume linearity between GPP and PAR.

**9)** Please note the typo in the legend of Figure 3.

Done.
* * *
**Reviewer #3 Comments:**

The authors take TROPOMI SIF retrievals from 2018 and 2019 and correlate them with Ameriflux GPP estimates to calculate a relationship between SIF and GPP. They then downscale this relationship to 500m resolution using NIRV to calculate GPP over CONUS using a 16-day moving window. They find a small difference in total CONUS GPP between 2018 and 2019 (4%), and determine that 28% of this variability can be explained by 4 precipitation-associated events, in Texas, the Midwest, South Dakota, and California.

We appreciate the reviewers comments and feedback.

**Comments**

**1.)  *There is nothing explicitly wrong with this paper. The methods are sound, and the results are reasonably explained by the data.* That** being said, my overall impression of the paper is that there is nothing novel or new here. The linear relationship (over large spatiotemporal scales) between SIF and GPP has been in the literature for a decade. That we can also see it in TROPOMI is not a dramatic finding. Previous work (e.g. MODSIF) has also downscaled SIF data. We also know that GPP is suppressed in drought. Again, a whole body of work. Vegetation response to drought as expressed by reduced SIF has also been previously reported. The suppression of early-season GPP due to Midwest flooding was in a paper last year. Using a new dataset to confirm previous results is not exactly groundbreaking. After reading the paper, my reaction was "well, yeah." Nothing new here, sort of a 'me too' paper, confirming previous results with a new dataset.
…
I can't in good conscience recommend a paper for publication just because they didn't do anything 'wrong'. I think there is a responsibility on the authors' part to present new, innovative and useful information to the community when submitting a paper, and I don't think that bar has been cleared here.
…
The title of the paper states that extreme events drive year-to-year variability in GPP. Between 2018 and 2019, the amount of IAV explained by the 'extreme' events emphasized is just a little over 25%. What about the other 75%? Are there other events that were not included in the study and would therefore increase the fraction of variability due to extreme events? Why were they not included? Or is the majority of IAV explained by smaller anomalies (that don't qualify as extreme) in precipitation, humidity and/or temperature? If the latter, then the paper title is demonstrably false. The claims made are qualitative, and after reading the paper I was not convinced that the title was a true statement.
…
I find myself wondering about the word 'extreme'. It's in the title (and not defined in the paper), so let's think about it a little bit. How might one define 'extreme event'? Is there a quantitative metric? I'm thinking about drought specifically, since 3 of the 4 'extreme' events involved precipitation deficit. There are multiple products that can be used to define drought (or high precipitation that might lead to flooding), such as the University of Nebraska-Lincoln Drought Monitor, Standardized Precipitation Index, or Palmer Drought Severity Index that are easily available in gridded form. It would not be difficult to define 'extreme' using one (or more) of these products and then correlate the regional differences in GPP with them. The title of the paper could then be confirmed or refuted quantitatively.

We appreciate the reviewer's positive assessment of the content and lack specific criticisms.

That said, we find the reviewer's assessment odd. While this reviewer does not seem to find the manuscript particularly interesting we contend that our manuscript is useful to the scientific community as it stands. Our manuscript presents a regional analysis of a novel dataset. We then identify the major anomalies that drive year-to-year variability in our dataset. We have also added a comparison to other independent GPP datasets (FLUXCOM and FluxSat) as well as MODIS NIRv and LAI. Further, Reviewer #1 and Reviewer #2 both specifically note that the work is interesting. As such, we argue that our work is a valuable contribution to the scientific literature and appropriately placed in a disciplinary journal such as Biogeosiences.

Regarding the Reviewer's comment *"The title of the paper states that extreme events drive year-to-year variability in GPP… Or is the majority of IAV explained by smaller anomalies (that don't qualify as extreme) in precipitation, humidity and/or temperature? If the latter, then the paper title is demonstrably false."*

We were unable to find a simple explanation (e.g., precip or humidity) for the remaining variability. Our Figure 3 shows a handful of large, obvious anomalies. Our work looked at the largest of these and found they were consistently driven by extreme precipitation events, thus the title of the paper.

*2)* 2018-2019 differences in GPP could be stratified by drought/excess precipitation metric, and the annual difference in GPP could be quantified in terms of fractional contribution by 'extreme' (greater than +/- 1 standard deviation in the index chosen?) and fractional contribution by anomalies that don't qualify as extreme. I'd be interested to see that. One might also look at these differences on a seasonal and/or PFT-level basis. Does 'minor' (not extreme) variability in eastern CONUS drive overall CONUS variability because mean GPP is so much larger than in more arid regions (the west)? That's an interesting question as well.

We appreciate the suggestion of the reviewer and think this may be interesting to investigate in future work. However, we don't think an additional exploratory analysis is warranted for this particular manuscript.

*3)* My formal recommendation for the paper is rejection, but I think with a little more effort the datasets the authors have produced can be used for research that has real value. If that effort can be put forth, I think resubmission would be entirely appropriate.

There were 4 main results from this paper: 1) we observe distinct relationships across different ecosystems between TROPOMI SIF and GPP, 2) we use these relationships to estimate GPP across CONUS at 500-m spatial resolution, 3) we observe small year-to-year differences in CONUS GPP (less than 4%), and 4) of the differences we do see, a large fraction of those are attributable to extreme precipitation events. All four of these results are the first of their kind.

Following suggestions from other reviewers, we have now added an additional comparison to FLUXCOM and FluxSat.

**Comments**

*1)* Page 4, L30-31: "…by assuming that GPP scales linearly with PAR…" This statement is just plain wrong. There is a very large body of observational work that demonstrates it. The idea that the GPP/PAR relationship is nonlinear is the basis of every terrestrial biosphere model, from light-response models like VPRM or CASA to enzyme-kinetic models such as CLM, ORCHIDEE or SiB. The authors of the paper know this, so I'm assuming that there are assumptions behind this statement. They need to be explained, and the statement justified (much) more than just a statement of fact. It is not.

We have made a slight change to the way the SIF:GPP relationship is inferred that should address this. We now compute a daily corrected SIF (see, for example, Frankenberg et al., 2011; Koehler et al., 2018) that corrects for variations in overpass time, length of day, and SZA. We then compare this daily corrected SIF with the daily integrated GPP at each AmeriFlux site. The relationship is now between daily corrected SIF and daily integrated SIF. As such, we no longer need to assume linearity between GPP and PAR.

*2)* There are multiple references that are incorrectly formatted.

Corrected.

*3)* The annual GPP was given (0.6-0.7 PG C), but comparison to other products would be helpful. It is well-known that GPP simulations (global or regional) vary by a factor of two or more, so it would be informative to know where the GPP product generated here sits in that spectrum, not just in terms of annual total but seasonal/regional comparison as well.

We now compare to FLUXCOM and FluxSat GPP as well.

[revised manuscript text omitted]